# Plasma Oxalate as a Predictor of Kidney Function Decline in a Primary Hyperoxaluria Cohort

**DOI:** 10.3390/ijms21103608

**Published:** 2020-05-20

**Authors:** Ronak Jagdeep Shah, Lisa E. Vaughan, Felicity T. Enders, Dawn S. Milliner, John C. Lieske

**Affiliations:** 1Division of Nephrology and Hypertension, Mayo Clinic, Rochester, MN 55905, USA; dr.ronakjagshah@gmail.com (R.J.S.); Milliner.Dawn@mayo.edu (D.S.M.); 2Division of Biomedical Statistics and Informatics, Mayo Clinic, Rochester, MN 55905, USA; Vaughan.Lisa@mayo.edu (L.E.V.); Enders.Felicity@mayo.edu (F.T.E.); 3Division of Pediatric Nephrology, Mayo Clinic, Rochester, MN 55905, USA; 4Division of Laboratory Medicine and Pathology, Mayo Clinic, Rochester, MN 55905, USA

**Keywords:** plasma oxalate, primary hyperoxaluria, estimated glomerular filtration rate, chronic kidney disease, Urine Oxalate, end-stage renal disease

## Abstract

This retrospective analysis investigated plasma oxalate (POx) as a potential predictor of end-stage kidney disease (ESKD) among primary hyperoxaluria (PH) patients. PH patients with type 1, 2, and 3, age 2 or older, were identified in the Rare Kidney Stone Consortium (RKSC) PH Registry. Since POx increased with falling estimated glomerular filtration rate (eGFR), patients were stratified by chronic kidney disease (CKD) subgroups (stages 1, 2, 3a, and 3b). POx values were categorized into quartiles for analysis. Hazard ratios (HRs) and 95% confidence intervals (95% CIs) for risk of ESKD were estimated using the Cox proportional hazards model with a time-dependent covariate. There were 118 patients in the CKD1 group (nine ESKD events during follow-up), 135 in the CKD 2 (29 events), 72 in CKD3a (34 events), and 45 patients in CKD 3b (31 events). During follow-up, POx Q4 was a significant predictor of ESKD compared to Q1 across CKD2 (HR 14.2, 95% CI 1.8–115), 3a (HR 13.7, 95% CI 3.0–62), and 3b stages (HR 5.2, 95% CI 1.1–25), *p* < 0.05 for all. Within each POx quartile, the ESKD rate was higher in Q4 compared to Q1–Q3. In conclusion, among patients with PH, higher POx concentration was a risk factor for ESKD, particularly in advanced CKD stages.

## 1. Introduction

Primary hyperoxaluria (PH) is a rare inherited autosomal recessive genetic disease caused by defects in genes that encode proteins important for glyoxylate metabolism [1]. Currently, three distinct forms are known—PH1 results from mutations in the enzyme alanine-glyoxylate aminotransferase (AGT) which is encoded the *AGXT* gene, PH2 is caused by a deficiency of the glyoxylate reductase/hydroxypyruvate reductase (GRHPR) enzyme encoded by *GRHPR*, and PH3 occurs when the mitochondrial enzyme, 4-hydroxy-2-oxoglutarate aldolase (HOGA) is deficient due to mutations in the *HOGA1* gene. Based upon current numbers in the Rare Kidney Stone Consortium (RKSC) PH registry, approximately 70% of diagnosed patients are PH1, 10% are PH2, 10% PH3, and 10% do not have an identified genetic cause [2].

The metabolic consequence of each of these enzyme deficiencies is a marked increase in hepatic oxalate production. Since oxalate cannot be metabolized by humans, the excess released into the plasma must be excreted by the kidneys, with less than 10% eliminated through the gastronintestinal tract. Calcium oxalate stones and nephrocalcinosis can result from high urinary oxalate (UOx) excretion; the latter can be associated with interstitial inflammation and fibrosis and may contribute to progressive chronic kidney disease (CKD) and end-stage kidney disease (ESKD) [3]. Once patients approach ESKD, excess oxalate can no longer be eliminated by the kidneys, and it accumulates in the body, potentially leading to systemic oxalosis. Among PH patients, there is wide variability in clinical course, with some progressing to ESKD in early childhood, while other PH patients retain kidney function into their fifth or sixth decade. Prediction of long-term outcomes using biomarkers is an important tool for clinical management, particularly now that novel treatment strategies with the potential to reduce hepatic oxalate production in PH are ready for clinical trials [4]. Patients with PH typically excrete >0.7 mmol/1.73 m^2^/day [5], and we previously reported that higher UOx predicts future ESKD risk within the PH patient group [2]. We also recently reported that plasma oxalate (POx) concentration correlates with UOx excretion [6].

Since 24 h urine collections can be difficult to obtain, especially on a repeated basis or in younger children, a blood biomarker that predicts UOx and other clinical features of PH could be clinically valuable. Furthermore, in patients with advanced CKD, UOx may no longer reflect systemic oxalate burden; in such cases, POx may represent a more accurate biomarker [7]. Therefore, we examined data in the RKSC PH registry in order to determine whether POx represents a viable biomarker that predicts the future loss of kidney function among patients with confirmed PH and at varying CKD stages.

## 2. Results

### 2.1. Baseline Characteristics

There were 227 patients who met the criteria for this study (Figure 1). During follow-up, ESKD developed in nine of the 118 patients (7.6%) in the CKD 1 group, 29 of 135 (21.5%) of the CKD 2 patients, 34 of 72 (47.2%) in the CKD3a, and 31 of 45 (68.8%) in the CKD 3b. There was one death in the CKD 1 group, nine deaths each in the CKD 2 and 3a groups, and 10 deaths in the CKD 3b group. The proportion of patients with PH1 increased by CKD stage (Table 1), representing 56.8% with CKD1, 73.3% with CKD2, 86.1% with CKD3a, and 84.4% with CKD3b. Due to the analysis plan, the median age at PH diagnosis also differed according to which patients experienced each CKD stage, from 5.4 years (CKD1) to 16.0 years (CKD3b). Median follow-up time was 5.3, 8.8, 6.6, and 1.8 years for CKD stages 1–3b, respectively.

Baseline POx increased by CKD stage from (3.1 (2.1, 5.7) µmol/L in CKD1 (*n* = 38) to 14.4 (10.5, 20.0) µmol/L in CKD3b (*n* = 17) (Table 2). The results were similar within the PH1 subset, increasing from 3.9 [2.4, 6.8] µmol/L in CKD1 (*n* = 24) to 14.9 (11.6, 21.5) µmol/L in CKD3b (*n* = 16). The numbers were not sufficient for a similar sub-analysis in PH2 and PH3. The risk of incident ESKD was higher in patients with PH1 compared to PH2 and PH3 in CKD2 and CKD3b; the results were similar albeit non-significant in CKD1 (HR 7.45; 95% CI 0.92–60.2; *p* = 0.06) and CKD3a (HR 5.74; 95% CI 0.78–42.1; *p* = 0.085) (Table 3).

### 2.2. POx and ESKD

When treated as a continuous predictor, higher baseline POx values were significantly associated with a higher risk of ESKD in CKD2 (HR 1.17; 95% CI 1.01–1.35; *p* = 0.033), CKD3a (HR 1.29; 95% CI 1.09–1.53; *p* = 0.004) and CKD3b (HR 1.24; 95% CI 1.08–1.42; *p* = 0.003) (Table 3). Baseline POx values in Q4 compared to POx in Q1 were also associated with a higher risk of ESKD in CKD3a (HR 13.88; 95% CI 1.41–137; *p* = 0.024) and CKD3b (HR 42.1; 95% CI 3.29–539; *p* = 0.004) (Table 3).

During follow-up (Table 4), POx was significantly associated with ESKD risk across all CKD stages: CKD1 (HR 1.12; 95% CI 1.02–1.24, *p* = 0.018), CKD2 (HR 1.17; 95% CI 1.08–1.25; *p* < 0.001), CKD3a (HR 1.19; 95% CI 1.11–1.27; *p* < 0.001), and CKD3b (HR 1.12; 95% CI 1.04–1.21; *p* = 0.003). When POx was considered by quartile, Q4 was a significant predictor of ESKD compared to Q1 across the CKD stages as well: CKD 2 (HR 14.2; 95% CI 1.76–115; *p* = 0.013), CKD3a (HR 13.7; 95% CI 3.02–62; *p* < 0.001), and CKD3b (HR 5.19; 95% CI 1.10–24.5; *p* = 0.038). Within each POx quartile, the ESKD rate was higher for the later CKD stages. Within each CKD stage, the ESKD rate was also higher in the fourth POx quartile compared to the first three (Appendix A and Figure 2). Thus, the greatest ESKD rate was for the CKD 3b subjects in the highest POx quartile (Figure 2). 

### 2.3. UOX and ESKD

Risk of ESKD increased at higher UOx levels when follow-up UOx was employed as a continuous time-dependent covariate, yielding an HR of 1.8 (95% CI 1.35–2.5) per each UOx increased of 1 mmol/1.73 m^2^/24 h (*p* < 0.001). When examined by follow-up UOx quartile (cut-off points of 0.77, 1.21, and 1.84 mmol/1.73 m^2^/24 h), UOx Q4 had a higher ESKD risk than Q1 (HR, 3.7; 95% CI 1.5–9.55) (*p* < 0.01).

### 2.4. ESKD or 40% Sustained Reduction in eGFR

Results were similar when the combined endpoint of ESKD and a 40% sustained eGFR reduction in eGFR were used for CKD progression. A higher baseline POx remained a significant predictor of incident CKD progression at CKD3b (HR 1.27; 95% CI 1.09–1.48; *p* = 0.002). Higher POx values during follow-up also remained a significant predictor of CKD progression in CKD2 (HR 1.15, 95% CI 1.07–1.23; *p* < 0.001) and CKD3a (HR 1.17, 95% CI 1.08–1.27; *p* < 0.001) when treated both as a continuous predictor and when comparing POx Q4 to POx Q1 (HR 10.71; 95% CI 1.34–85.4; *p* = 0.025 and HR 8.15; 95% CI 1.74–38.2; *p* = 0.008; respectively). Follow-up time was shorter, and there were fewer laboratory parameters available when considering this composite endpoint. 

### 2.5. eGFR Slope

There were 59 patients with a total of 369 POx and eGFR laboratory measures obtained within three months of each other throughout follow-up. The number of lab values per patient ranged from 1 to 20. After adjusting for follow-up time, eGFR was significantly lower among those with higher POx (eGFR reduced by 1.27 mL/min/1.73 m^2^ per 1 µmol/L increase in POx; (*p* < 0.001).

## 3. Discussion

In the current study, we analyzed the predictive value of POx for the subsequent decline in eGFR in PH patients, stratified by CKD stage. These data suggest that POx is a useful predictor of ESKD risk across CKD stages 2–3b (Figure 2), with the effect most pronounced in CKD3b. These data and our previous study [2] suggest that the use of POx and UOx could be complimentary, with UOx being particularly informative across CKD stages 1–3b, and POx particularly informative in CKD stages 3a and 3b. 

The clinical management of PH is challenging due to the lifelong nature of the disease and the risk for ESKD observed in a vast majority of PH1 patients, although this can occur at markedly variable ages. The ability to predict long term outcomes using biomarkers facilitates the most effective use of current treatments and also has the potential to provide an important outcome measure for clinical trials of novel therapeutics. Newer treatment options, including the potential use of small inhibitory RNA (siRNA) therapeutics to impact oxalate generating pathways in the liver are under current development [8]. These newer approaches make it important to better understand the prognostic features of PH, which could both identify patients who could be eligible candidates for clinical trials and also potentially be used as surrogate endpoints in future studies. 

In the current study, we examined whether POx is a useful prognostic marker for the future of loss of kidney function. The data demonstrate that higher POx levels both at baseline and during follow-up were significantly associated with loss of kidney function over time. When stratified by quartile, those in POx Q4 were at increased risk of ESKD compared to POx Q1 across CKD stages 2–3b. As expected, based upon previous prevalence data, patients that progressed to later CKD stages tended to be older and more likely to have PH1. The current study also confirms our previous observation that baseline UOx and UOx over follow-up predict ESKD, with those in the highest quartile at the greatest additional risk [2]. Furthermore, the HRs for the subsequent ESKD of the highest UOx quartile at baseline (2.5) and during follow-up (3.7) were similar to our previous work [2]. 

Much as the serum creatinine concentration is a net result of creatinine generation from muscle and elimination by the kidneys, POx is the net result of oxalate generated in the liver and absorbed from the GI tract and its elimination by the kidneys. Thus, higher POx can reflect higher hepatic oxalate generation, greater gastrointestinal absorption, lower GFR, or some combination of these. Indeed, our previous publications demonstrated that UOx excretion could be used to predict POx and eGFR [6]. Recent studies also suggest that higher UOx excretion predicts a higher renal tubular fluid oxalate concentration at the S3 segment of the proximal tubule, the anatomic site of nephrocalcinosis in PH [9]. Thus, the results from our current study demonstrating the predictive value of POx across the CKD stage 2–3b spectrum may reflect the fact that POx provides integration of oxalate generation and elimination, and thus becomes a sensitive marker of oxalate burden at the level of the proximal tubule.

One issue with the widespread use of POx is the challenging nature of the available laboratory assays [10]. Under normal circumstances, POx is present in micro-molar concentrations in blood. Thus, sample handling, including prevention of ascorbate conversion to oxalate, is quite important. In addition, methods for measuring oxalate, including sample type, preparation, and analysis, are not interchangeable [10,11,12]. Our study benefitted from use of a single clinical laboratory over many decades with data to support that POx results could be compared over that time period. However, because of the barriers for analysis in routine laboratories, the POx measurement is not widely available. Additionally, when GFR is normal or near-normal, POx concentrations typically are near or below the limit of quantification in the general population [10]. However, due to a markedly increased oxalate generation, POx is typically above the limit of quantification in most PH patients, even with preserved eGFR [6], suggesting that with a consistent and sensitive assay, POx could be a useful biomarker. Moreover, as GFR declines, POx values rise well within the quantification range with good reproducibility in all PH patients. 

The current study has several limitations. Due to the retrospective nature of this analysis based on registry data, laboratory measures and follow-up were limited by availability, and variability in the diagnosis and ability to recruit patients with this rare disease may have introduced bias. There was also no controlled intervention to change POx values. Nevertheless, we were able to analyze a relatively large cohort of over 200 PH patients with available POx values. Furthermore, comorbidities such as obesity, dyslipidemia, hypertension, and albuminuria, which are important CKD risk factors, were not available for multivariate analysis. However, many PH patients progress to ESKD at a very young age, and thus these comorbidities likely play a relatively minimal role in CKD progression, in comparison to common causes of CKD such as diabetic nephropathy in which vascular/microvascular injury and dysfunction play a major role.

In conclusion, there is a need for improved knowledge regarding the utility of biomarkers to predict ESKD risk in PH patients at various stages of CKD. The current study suggests that POx, perhaps in combination with other risk factors, is a useful marker for this purpose. 

## 4. Materials and Methods

Natural history and laboratory data from PH patients enrolled in the RKSC PH Registry were used for analysis [13]. PH1, PH2, and PH3 patients were confirmed by mutations in the AGXT, GRHPR, or HOGA1 genes, respectively. POx was measured in the Mayo Clinic Renal Testing Laboratory (Rochester, MN, USA) by an oxalate oxidase (1991–6/2016) or ion chromatography (6/2016–2019) based-assay per the standard Mayo Renal Testing Laboratory Protocol [10]. Detailed validation data was available to determine that assay results could be compared over the course of this time period. UOx was measured by oxalate oxidase also in the Mayo Renal Testing Laboratory or another accredited clinical laboratory [10,14]. Data for subjects <2 years old were excluded from this analysis due to potential confounding effects of renal maturation in very young children on GFR (and thus POx). 

This project was approved by the Mayo Clinic Institutional Review Board (IRB 11-001702; initial approval 16 August 2011). A total of 545 PH patients were identified in the Registry as of March 31, 2019 (Figure 1). After excluding patients who met clinical criteria but had no detectable mutations of AGXT, GRHPR, or HOGA1, (*n* = 46), those with ESKD at diagnosis (*n* = 144), patients less than two years old at ESKD or last follow-up (*n* = 22), and those patients without eGFR data after diagnosis and older than two years before ESKD or death (*n* = 106), a total of 227 patients remained in the final cohort for analysis. Since our previous study suggested an association between POx and CKD stage [6], patients were then divided into four groups based on CKD stage (1, 2, 3a, 3b) in a landmark-style analysis, such that a patient started in a given CKD stage subgroup on their first eGFR observed in that range, while also remaining in any prior groups through all available follow-ups. For instance, a patient whose eGFR measurements were 64, 72, 43, 47, 38, 29, and 21 mL/min/1.73 m^2^ would never be included in the CKD stage 1 group, would enter the eGFR 60–89 group with the date of the eGFR = 64 mL/min/1.73 m^2^ measurement and be followed until kidney progression or censoring, would enter the eGFR 30–44 group with the date of the eGFR = 43 mL/min/1.73 m^2^ measurement and be followed until kidney progression or censoring, and would enter the eGFR 45–59 group with the date of the eGFR = 47 mL/min/1.73 m^2^ measurement and be followed until kidney progression or censoring.

Laboratory results were extracted from the registry data, and were from baseline (defined as within one year prior to or within six months of entry into the CKD stage group and prior to kidney progression) or follow-up (defined from six months after the entry into the CKD stage group and prior to kidney progression). None of this patient cohort experienced acute kidney injury events during follow-up. GFR was estimated via the CKD-EPI creatinine equation for adults greater than 18 years old [15] and the Schwartz equation for those less than 18 years old [16]. POx values were manually examined, and outlying transient results not fitting the clinical picture were removed from the analysis.

### 4.1. Statistical Methods

#### Plasma Oxalate 

Progression to ESKD (eGFR < 15 mL/min/1.73 m^2^ or the start of dialysis or renal transplantation) was selected as the primary endpoint. Sensitivity analyses were also performed using a combined endpoint of ESKD or sustained 40% reduction in eGFR from baseline. The results were expressed in terms of the median (25th, 75th percentiles) for continuous variables and as percentages for categorical variables. To maximize available data, the diagnosis/baseline labs were defined as the closest reading between one year before entry and up to six months after entry into a CKD stage group. 

The percentage of patients who were free of renal progression (ESKD and > 40% decline in eGFR) after entry into each CKD stage was estimated using the Kaplan–Meier method. The effects of baseline clinical characteristics, as well as Pox on renal progression, were estimated by univariate analyses using the Cox proportional hazard model with log-rank tests. The primary outcome of interest was time to ESKD and was censored on death or loss to follow-up. Hazard ratios (HRs) and 95% confidence intervals (95% CIs) are presented. A time-dependent Cox model was used to explore the effect of POx concentration on renal outcome during follow-up. Times to ESKD by POx quartile during follow-up were estimated for each CKD stage group by dividing individual-patient follow-up time into intervals based on the time between POx measures or last follow-up. Person-time and ESKD events were summed within the POx quartile with the rate = 100 × (Events/Person-time). The proportional hazards assumption was checked for all models using martingale residuals. Generalized estimating equations (GEE) adjusting for time were used to evaluate the association between POx and eGFR throughout follow-up. The effect of follow-up UOx concentration on the risk of renal outcomes were also assessed using a time-dependent Cox model.

## Figures and Tables

**Figure 1 ijms-21-03608-f001:**
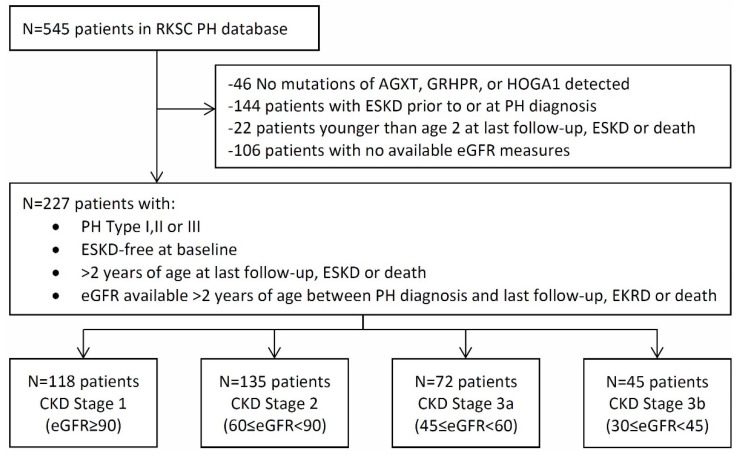
Flowchart of inclusion criteria for analysis cohort. From a total of 545 PH1 patients in the Registry, 227 were eligible for this analysis.

**Figure 2 ijms-21-03608-f002:**
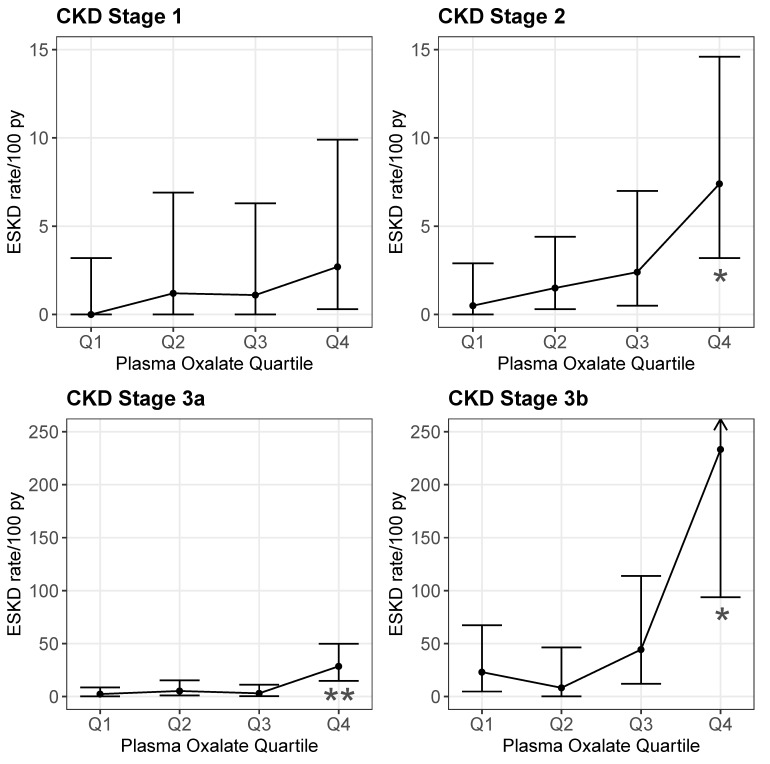
ESKD rate by POx quartile during follow-up by CKD stage. ESKD rates were estimated for each CKD stage group by dividing individual patient follow-up times into intervals based on the time between the POx measures or last follow-up. Person-time and ESKD events were summed within POx quartiles with the rate = [100 × (events/person-time)]; error bars represent 95% CI of the ESKD rate (see Appendix A for numerical values). ESKD rates were similar for the lower three quartiles (Q) but increased for the highest POx quartile across CKD stages 2–3b (Q4 vs. Q1 HR 14.21; 95% CI 1.76–114.7; * *p* < 0.05 for CKD stage 2, HR 13.66; 95% CI 3.02–61.91; ** *p* < 0.001 for CKD stage 3a, and HR 5.19; 95% CI 1.10–24.5; * *p* < 0.05 for CKD stage 3b).

**Table 1 ijms-21-03608-t001:** Clinical characteristics of patients with primary hyperoxaluria who did not have ESKD at or before diagnosis.

	CKD Stage			
	Stage 1 (≥90)	Stage 2 (60–89)	Stage 3a (45–59)	Stage 3b (30–44)
**Characteristics**	***n* = 118**	***n* = 135**	***n* = 72**	***n* = 45**
**Type of PH, % (*n*)**				
PH1	67 (56.8%)	99 (73.3%)	62 (86.1%)	38 (84.4%)
PH2	23 (19.5%)	16 (11.9%)	6 (8.3%)	5 (11.1%)
PH3	28 (23.7%)	20 (14.8%)	4 (5.6%)	2 (4.4%)
**Sex, % (*n*)**				
Male	68 (57.6%)	76 (56.3%)	39 (54.2%)	23 (51.1%)
Female	50 (42.4%)	59 (43.7%)	33 (45.8%)	22 (48.9%)
**Age at diagnosis, y**	5.4 (2.7, 11.1)	7.9 (4.0, 23.9)	10.7 (4.6, 26.4)	16.0 (7.0, 41.7)
**Follow-up time, y**	5.3 (2.9, 10.0)	8.8 (3.1, 15.2)	6.6 (3.4, 12.8)	1.8 (1.0, 3.8)
**Patients with follow-up Pox labs**	***n* = 50**	***n* = 69**	***n* = 37**	***n* = 17**
**No. follow-up labs**	2.5 (1,5)	3 (1,6)	3 (1,5)	1 (1,3)

Continuous variables are expressed as median with 25th, 75th percentiles. *n*, number; PH, primary hyperoxaluria; PH1, primary hyperoxaluria type 1; PH2, primary hyperoxaluria type 2; PH3, primary hyperoxaluria type 3; y, years.

**Table 2 ijms-21-03608-t002:** Baseline and follow-up POx quartiles, by CKD stage.

Oxalate Measure	*n*	Q1	Median	Q3
**Baseline**				
**POx, umol/L**				
CKD stage 1	38	2.1	3.1	5.7
CKD stage 2	44	1.9	4.1	7.2
CKD stage 3a	25	2.9	4.8	8.2
CKD stage 3b	17	10.5	14.4	20.0
**Follow-up**				
**POx, umol/L**				
CKD stage 1	171	1.9	3.0	4.8
CKD stage 2	288	2.1	4.1	7.1
CKD stage 3a	165	4.2	7.0	12.9
CKD stage 3b	38	9.9	15.2	18.0

**Table 3 ijms-21-03608-t003:** Factors univariately associated with incident ESKD among patients with primary hyperoxaluria without ESKD at baseline.

	CKD Stage															
	Stage 1 (≥90)				Stage 2 (60–89)				Stage 3a (45–59)				Stage 3b (30–44)			
**Variable**	***n*; E**	**HR (95% CI)**	***p***	**C-Index**	***n*; E**	**HR (95% CI)**	***p***	**C-Index**	***n*; E**	**HR (95% CI)**	***p***	**C-Index**	***n*; E**	**HR (95% CI)**	***p***	**C-Index**
**Demographics**																
**PH1**	118; 9	7.45 (0.92–60.2)	0.06	0.677	135; 29	23.9 (3.06–186)	**0.003**	0.630	72; 34	5.74 (0.78–42.1)	0.085	0.544	45; 31	9.45 (1.28–69.8)	**0.028**	0.607
**Male**	118; 9	1.31 (0.32–5.34)	0.71	0.535	135; 29	1.73 (0.80–3.74)	0.17	0.577	72; 34	1.10 (0.56–2.18)	0.78	0.517	45; 31	1.07 (0.51–2.23)	0.86	0.520
**Age at diagnosis**	118; 9	0.98 (0.90–1.08)	0.69	0.500	135; 29	1.00 (0.98–1.03)	0.78	0.577	72; 34	1.01 (0.98–1.03)	0.70	0.550	45; 31	0.98 (0.96–1.01)	0.13	0.612
**Plasma Oxalate**																
**POx,** **umol/L**	38; 2	NE ^†^	NE ^†^	NE ^†^	44; 9	1.17 (1.01–1.35)	**0.033**	0.623	25; 10	1.29 (1.09–1.53)	**0.004**	0.727	17; 14	1.24 (1.08–1.42)	**0.003**	0.810
**POx, quartile**	38; 2	-	-	NE ^†^	44; 9	-	-	0.610	25; 10	-	-	0.698	17; 14	-	-	0.791
**Q1**	-	REF	REF	-	-	REF	REF	-	-	REF	REF	-	-	REF	REF	-
**Q2**	-	NE ^†^	NE ^†^	-	-	0.41 (0.04–4.52)	0.47	-	-	2.40 (0.14–40.0)	0.54	-	-	7.85 (0.82–75.0)	0.074	-
**Q3**	-	NE ^†^	NE ^†^	-	-	1.68 (0.27–10.52)	0.58	-	-	4.39 (0.49–39.5)	0.19	-	-	10.1 (1.11–91.5)	**0.040**	-
**Q4**	-	NE ^†^	NE ^†^	-	-	1.57 (0.25–9.75)	0.63	-	-	13.88 (1.41–136.5)	**0.024**	-	-	42.1 (3.29–539)	**0.004**	-

PH1, primary hyperoxaluria type 1; *n*: number available for analysis; E: = ESKD events; 95% CI, 95% confidence interval. *p*-values in bold denote significance at the 0.05 level. Harrell’s c index is provided. The proportional hazards assumption was met for all models with reported HRs. ^†^ Not estimable, sample size, and no. of events too small for a variable with four levels.

**Table 4 ijms-21-03608-t004:** Plasma oxalate excretion on follow-up (>6 months after entry into the CKD group) and risk of ESKD.

	CKD Stage															
	Stage 1 (≥90)				Stage 2 (60–89)				Stage 3a (45–59)				Stage 3b (30–44)			
**Follow-up Plasma Oxalate**	***n*; E**	**HR (95% CI)**	***p***	**C-Index**	***n*; E**	**HR (95% CI)**	***p***	**C-Index**	***n*; E**	**HR (95% CI)**	***p***	**C-Index**	***n*; E**	**HR (95% CI)**	***p***	**C-Index**
**POx, umol/L**	171; 4	1.12 (1.02–1.24)	**0.018**	0.854	288; 15	1.17 (1.08–1.25)	**<0.001**	0.806	165; 19	1.19 (1.11–1.27)	**<0.001**	0.795	38; 15	1.12 (1.04–1.21)	**0.003**	0.729
**POx, quartile**	171; 4	-	-	0.818	288; 15	-	-	0.772	165; 19	-	-	0.757	38; 15	-	-	0.752
**Q1**	-	REF	REF	-	-	REF	REF	-	-	REF	REF	-	-	REF	REF	-
**Q2**	-	NE ^†^	NE ^†^	-	-	2.70 (0.28–26.03)	0.39	-	-	2.10 (0.35–12.65)	0.42	-	-	0.39 (0.04–4.40)	0.45	-
**Q3**	-	NE ^†^	NE ^†^	-	-	3.98 (0.41–38.82)	0.24	-	-	1.28 (0.18–9.17)	0.80	-	-	0.98 (0.19–5.04)	0.98	-
**Q4**	-	NE ^†^	NE ^†^	-	-	14.21 (1.76–114.7)	**0.013**	-	-	13.66 (3.02–61.91)	**<0.001**	-	-	5.19 (1.10–24.5)	**0.038**	-

^†^ Not estimable, sample size, and no. events, too small for a variable with four levels. *p*-values in bold denote significance at the 0.05 level. Harrell’s c index is provided. *n* = Follow-up intervals; E = ESKD events.

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
