# Peer review of "Plasma Oxalate as a Predictor of Kidney Function Decline in a Primary Hyperoxaluria Cohort"

_ijms, 2020, doi:10.3390/ijms21103608_

Round 1
Reviewer 1 Report
The authors put an interest question, if the levels of oxalate in patients with PH could be related or not with the risk of CKD.
Nedless to say the the authors´s reponse was yes.
My mejor problem with the interpretarion of this fact is based solely in univariate analysis.
CKD and advanced CKD, even in PH is a complex condition with many aspects related to renal function loss: i.e. episodes of AKI, hypertension, obesity, cardiovascular disease (concomitant), albuminuria-proteinuria, dyslipidemia, medications, etc.
So, an univariate analysis is clearly insufficient to evaluate the independent effect of oxalate on disease progression.
Why this study was not done?
Also, according to figure 2 there is singnificant overlap between groups, leading to doubt about the independent role oxalate levels in CKD
Finally, the number of cases in each CKD stage is limited...and accordingly the HR for oxalate is quite low....les than 1.4. One would expect a more relevant role of blood oxalate in hyperoxaluria.....
My suggestion would be to abandon CKD stages (a cause of error) and spliting the database in several groups.....(another cause of bias)
I would do an linear regresion analysis with oxalate at baseline and eGFR at baseline as anotehr covariate, togetjer with other classical risk factor for renal disease progression. The outcome would be eGFR at follow-up
Author Response
The authors put an interest question, if the levels of oxalate in patients with PH could be related or not with the risk of CKD.
Nedless to say the the authors´s reponse was yes.
My mejor problem with the interpretarion of this fact is based solely in univariate analysis.
CKD and advanced CKD, even in PH is a complex condition with many aspects related to renal function loss: i.e. episodes of AKI, hypertension, obesity, cardiovascular disease (concomitant), albuminuria-proteinuria, dyslipidemia, medications, etc.
So, an univariate analysis is clearly insufficient to evaluate the independent effect of oxalate on disease progression.
Why this study was not done?
Thanks for these comments. We agree with the reviewer that when evaluating progression for most causes of CKD, comorbidities such as obesity, dyslipidemia, hypertension, and albuminuria are important risk factors as ultimately vascular/endothelial processes drive progression. The situation for PH differs in several important ways. Many patients progress to ESKD at a very young age and these comorbidities never play a role in renal or other organ dysfunction. More importantly, the mechanism of CKD progression is oxalate nephropathy/nephrocalcinosis driven by the underlying metabolic defect. This is quite different from other causes of EKKD, such as diabetic nephropathy or primary glomerular disorders.
The question of AKI is an interesting one IN PH, most patient progress quickly to ESKD if they do develop AKI, regardless of CKD stage. Thus, this would not be a meaningful predictor of CKD, as indeed most patients have an event of AKI at the time of ESKD onset. One could consider a study looking at AKI as an outcome, and that will be an interesting future analysis.
Reviewer 2 Report
This interesting study examines plasma oxalate at the various stages of CKD and how measurement of POx can be used as a biomarker for changes toward further loss of kidney function which is not dependent on the patients age.
I have the following minor comments;
- Line 161 "Recent studies also suggest that higher UOx excretion reflects predicts..." Delete one of the last 2 words or correct the syntax
- Throughout the manuscript the abbreviation for plasma oxalate is POx but in all of the tables it is Pox, POx should be used in the tables
- Additional discussion of figure 2 would be helpful for the reader. Also error bars are shown but the values are not.
- Also in the legend of Figure 2 a supplementary table 1 is mentioned. This reviewer did not have access to this table so no comment can be made on whether it should be included.
Author Response
This interesting study examines plasma oxalate at the various stages of CKD and how measurement of POx can be used as a biomarker for changes toward further loss of kidney function which is not dependent on the patients age.
I have the following minor comments;
- Line 161 "Recent studies also suggest that higher UOx excretion reflects predicts..." Delete one of the last 2 words or correct the syntax
“reflects” deleted.
- Throughout the manuscript the abbreviation for plasma oxalate is POx but in all of the tables it is Pox, POx should be used in the tables
We have changed to POx in the tables.
- Additional discussion of figure 2 would be helpful for the reader. Also error bars are shown but the values are not.
Thanks. The error bars represent the 95% CI of the ESKD event rates, now mentioned in the Table legend. The numerical values are in the now uploaded supplemental table 1. We have also added the sentence “Thus the greatest ESKD rate was for CKD 3b subjects in the highest POx quartile (Figure 2).” to the revised manuscript.
.
- Also in the legend of Figure 2 a supplementary table 1 is mentioned. This reviewer did not have access to this table so no comment can be made on whether it should be included.
Supplemental Table 1 has now been uploaded, which is a numerical representation of the data in Figure 2.
Reviewer 3 Report
The authors proposed a retrospective analysis which investigated plasma oxalate (POx) as a potential predictor of end stage kidney disease (ESKD) among primary hyperoxaluria patients. This research is a novel design, and execution and analysis are quite complete. Only a few details need to be clarified.
Major comments
- This study considered the age, gender and CKD stage as confounders for estimated the effect of primary hyperoxaluria. However, there were still some important confounders, such as hypertension and anemia, for the risk of ESKD in pediatric patients. They should be included the analysis model or discussed their impact.
- Although the authors set death as a censor in this study, it was interesting the death numbers in each CKD stage.
- The results showed the higher POx level was associated with loss of kidney function over time. It was curious that the ability of predictors of this model. The Harrell’s c index should also be reported by CKD stages in the research.
Minor remarks
- Several typos: line 155, “an d” should be revised to “and”. Moreover, all eGFR unit were ml/min/1.73 m2 in your manuscript (line 51,132, 205,207,208, 210) should be revised to ml/min/1.73 m2. Similarly, the units of “mmol/1.73 m2/24 hrs” should be revised to “mmol/1.73 m2/24 hrs” (line 116, 118)
Author Response
The authors proposed a retrospective analysis which investigated plasma oxalate (POx) as a potential predictor of end stage kidney disease (ESKD) among primary hyperoxaluria patients. This research is a novel design, and execution and analysis are quite complete. Only a few details need to be clarified.
Major comments
- This study considered the age, gender and CKD stage as confounders for estimated the effect of primary hyperoxaluria. However, there were still some important confounders, such as hypertension and anemia, for the risk of ESKD in pediatric patients. They should be included the analysis model or discussed their impact.
Thank you for this comment. In our opinion, anemia would be a result and not cause of CKD. Hypertension is quite rare in PH, in part because they typically progress to CKD at a relatively young age. Calcium oxalate infiltration of the vasculature may also be a factor, once advanced CKD occurs.
- Although the authors set death as a censor in this study, it was interesting the death numbers in each CKD stage.
Agreed. It is a disease with high mortality when ESKD occurs. We have now added the numbers of those who died during follow-up in each CKD stage in the results section.
- The results showed the higher POx level was associated with loss of kidney function over time. It was curious that the ability of predictors of this model. The Harrell’s c index should also be reported by CKD stages in the research.
I have now added this statistic to the two tables.
Minor remarks
- Several typos: line 155, “an d” should be revised to “and”. Moreover, all eGFR unit were ml/min/1.73 m2 in your manuscript (line 51,132, 205,207,208, 210) should be revised to ml/min/1.73 m2. Similarly, the units of “mmol/1.73 m2/24 hrs” should be revised to “mmol/1.73 m2/24 hrs” (line 116, 118)
Thank you; these have been corrected.

Round 2
Reviewer 1 Report
The mayor limitation that has the study, which is the lack of multivariate analysis, has not changed.
CKD, is generated in this case by oxalate, but then, there are other aspects relevant that may be related to renal function loss.
You are not taken into account other factors. Are we sure that oxalate levels is the only factor incolved in renal function loss? By this study you gice the impression that oxalate is the one and only factor involved.
Also, there is a major overlap in oxalate levels in the figures.....how do you explain this aspect in the relationship between the levels of oxalate and CKD progression?
Author Response
X
